

# Anoxia and salinity changes: a new Permian catastrophe record

Marlise C. Cassel[1], Ernesto L. C. Lavina[1], Joice Cagliari[1], René Rodrigues[2], Egberto Pereira[2]

[1]Universidade do Vale do Rio dos Sinos, Programa de Pós-Graduação em Geologia, Av. Unisinos, 950, Cristo Rei, São Leopoldo, Rio Grande do Sul, CEP 92022-000, Brazil
[2]Universidade do Estado do Rio de Janeiro, Departamento de Estratigrafia e Paleontologia, Rua São Francisco Xavier, 524, Maracanã, Rio de Janeiro, Rio de Janeiro, Brazil

*Correspondence to*: Marlise C. Cassel (marlise.cassel@gmail.com)

**Abstract.** Bituminous shales are associated to worldwide geological events, such as mass extinction, anoxia and climatic changes, mainly when preserved in carbonate ramps, and constitute reliable records of this dynamics. However, a minority of data still exist that measure the Permian, especially in the Southern Hemisphere associated to Panthalassic Ocean as compared to numerous studies of Oceanic Anoxic Events in the Cretaceous or associated to Tethys. The Late Permian extinction was the most severe extinction of the past 500 million years, which wiped out over 90% of marine species. Before the Permian-Triassic boundary (e. g. Early-Middle Permian successions) geochemical and geological anomalous aspects has received more attention due the information about the Late Paleozoic icehouse-greenhous transitions, which is one of the triggers for the catastrophic ectinction. Therefore the Irati Formation, Permian interval of Paraná Basin in southern Brazil, is suitable for the study of these intricate processes due to the presence of Permian bituminous shales and carbonates. Based on core descriptions, spectral gamma ray data and organic geochemistry, a stratigraphic scheme is here proposed to support paleoenvironmental inferences. We identified three depositional sequences formed internally by T-R cycles of highest frequency. The sedimentary facies analysis indicates a carbonate ramp subdivided into outer, middle and inner. Climatic and sea level changes are defined, and also oscillations in the salinity, oxygenation and organic matter source. Bituminous shales record normal salinity, with anoxia levels and even euxinia, associated to the increase in bioproductivity. Carbonate facies register periods of hypersalinity under oxic environment and semi-arid conditions. The accumulation of *Mesosaurus* skeletal remains results from the action of reworking fluxes, creating an endemic facies in this region. Climatic versus eustatic influence were differentiated, and also the controlling hierarchies. These results are evidence for the environmental dynamics from the Permian, that generated extreme global events, mainly in Gondwana.

## 1 Introduction

Black shales are associated to both regional and global geologic events, such as climatic changes, oceanic anoxia and mass extinction (Hotinski et al., 2001; Phelps et al., 2015). Over the Earth´s history, the anoxic events were concomitant with significant perturbations in the global carbon cycle, influencing the carbon dioxide ($CO_2$) concentration both in the atmosphere and in the oceans (Knoll et al., 1996; Grice et al., 2005; Bonis et al., 2010; Phelps et al., 2015). The most expressive records of these events are in carbonate platforms with occurrence of abrupt changes between carbonatic facies and anoxic shales (Knoll et al., 1996; Phelps et al., 2015).

Permian black shales are studied under the biogeochemical point of view, mainly because of the mass extinction at the end of this period (Knoll et al., 1996; Wignall and Twitchett, 1996; Hotinski et al., 2001; Zhang et al., 2001; Grice et al., 2005; Knoll et al., 2007; Clapham and Payne, 2011; Joachimski et al., 2012; Song et al., 2014; van Soelen et al., 2018). Some models propose a physiologic crisis resulting from the partial pressure of carbon dioxide ($pCO_2$) and temperature increase, associated to the partial pressure of oxygen ($pO_2$) reduction and carbonate saturation, which caused acidification and anoxia. Also, salinity changes are commonly associated to $pCO_2$ changes (Knoll et al., 2007; van Soelen et al., 2018). A recent focus to explain the Permian extinction is the coordinated imposition of physical perturbations that are linked (e. g. Bolide impact,





Siberian flood basalt volcanism, anoxia in the oceans and release of seafloor methane) (Knoll et al., 2007). Recently earlier successions has received more attention (e. g. Kungurian) (Liu et al., 2017) due geochemical and geological anomalus records that bring informations about Late Permian icehouse-greenhouse transition. This passage is one of the triggers for the catastrophic extinction. Some models predict that high-resolution stratigraphy within basin will show extinctions to be rapid

and pulsed. This was recurrent in other moments of Earth's history. In the Neoproterozoic these features occurred repeatedly over a 250 My interval (Knoll et al., 1996).

The worldwide aforementioned problem can be analyzed by a key case in the Southern Hemisphere. Several cratonic basins in South America also record permian black shales intervals that correspond to the same tecto-sedimentary cycle (Soares et al., 1974). This cycle of continental expression is synchronous with huge sedimentary units of the center-east of North

America, as proposed by Sloss (1963). In the Permian of the Paraná Basin (Figs. 1 and 2) and adjacent gondwanic basins of Chaco-Paraná (Argentina), Karoo and Kalahari (Southern Africa), thick packages of organic rich shales occur, intercalated with carbonatic layers (Oelofsen and Araújo, 1983; Oelofsen and Araújo, 1987; Hachiro, 1996; Milani et al., 1998; Araújo, 2001).

The Irati Formation is a key example to be studied. The formation was deposited during the Permian and is composed by

organic shales, siltstones and carbonates (Rocha-Campos et al., 2011; Santos et al., 2006). Therefore, this formation is considered the Permian record of environmental dynamics that generated extreme global events.

However, due to its broad area (extending for ~2000 km in the N-S axis) and faciologic variations associated to the outcropping area restricted shortly to the basin margin, the Irati Formation needs detailed studies and paleoenvironmental interpretations. Previous studies did not reach agreement over the paleoenvironment, two main debated topics being salinity

and its organic matter.

The main purpose of this work is the analysis of the paleoenvironmental significance from the southern hemisphere Permian black shales and carbonatic rocks, based on organic geochemistry and geophysical approaches. This study improves the understanding of both the climatic and the eustatic controls during high frequency sequences and a high-resolution stratigraphic scheme for the bituminous shales and carbonates of the Irati Formation. The results showing variations in the

organic matter composition, oxigenation and salinity are discussed as evidence from Permian environmental changes.

## 2 Methodology

Twenty-three cores from the southern part of Paraná Basin were described and analyzed using spectral gamma-ray (Fig. 3). The obtained facies scheme was analyzed in a sequence stratigraphy approach, according to Hunt and Tucker (1992) depositional model.

A total of 214 samples were collected and analyzed at the Laboratório de Estratigrafia Química e Geoquímica Orgânica da Faculdade de Geologia, Universidade Estadual do Rio de Janeiro. Two cores were sampled every 60 cm for total organic carbon (TOC, %), total sulfur (S, %) and insoluble residual ($R_{ins}$, %). The $R_{ins}$ (%) was obtained by hot acidification for elimination of carbonates. A LECO SC 632 equipment was used for both TOC (%) and S (%) analyses. Subsequently, 52 samples were chosen for pirolysis-rock eval analyses, which supplied hydrogen index (HI) and oxygen index (OI) values in

Rock Eval 6 Vinci Technologies.

Additionally, new gamma-ray measurements were acquired using a gamma spectrometer RS 124 Radiation Solutions, with three minute acquisition period every 50 cm. These new data allowed the differentiation of potassium (K, %), thorium (eTh, ppm) and uranium (eU, ppm) channels and the elaboration of new profiles, whose respective curves were normalized to improve the comparison of trends.

The cyclicity was observed in the total gamma-ray curve, whose radioactivity corresponds to the sum of K (%), eTh (ppm) and eU (ppm) (Glover, 2012). In the case of siliciclastic rocks of fine granulometry that represents the highest part of the



succession here described, the variations of K (%) are associated to the clay content of rocks, indicating variations of higher and lower energy in the environment due to oscillations in water depth (Milani, 1997; Kearey et al., 2009). Therefore, the increase in K (%) values corresponds to a transgressive trend. The eTh (ppm), in the form of resistates such as the detritic mineral monazite (Andersson and Worden, 2004; Kearey et al., 2009), presents negative covariation with K (%) indicating

higher energy intervals and increased sedimentary input. On the other hand, in the cases of positive covariation with K (%), the eTh (ppm) is associated to clay minerals by adsorption (Syed, 1999). In the case of eU (ppm), due to its behavior in relation to other radioactive elements, presenting negative covariation, it is considered authigenic, corresponding to low oxygenation contexts. With the availability of ion $U^{6+}$ soluble in seawater, in low oxygen levels occurs the reduction to $U^{4+}$ and precipitation as insoluble compounds of uranium dioxide ($UO_2$) or hydroxides ($U(OH)_4$) (Wingall and Twitchett, 1996;

Lünning and Kolonic, 2003; Bonotto and Silveira, 2006). A formula for the calculation of authigenic uranium from spectral gamma-ray takes the form $U_{authigenic} = U_{total} - Th/3$ (Wingall, 1994). Therefore ratios between these elements are also proxys. The ratio eTh/eU with values near or below 2 indicates anoxia, whereas values between 2 and 7 indicate oxyc marine environment and higher values indicate oxic continental input (Wingall and Twitchett, 1996).

## 3 Regional Geology

The Paraná Basin is an intracratonic basin that spreads through central and southern Brazil, northern Argentina, Paraguay and Uruguay, covering an area of 1 1,500,000 km² (Fig. 1). The Phanerozoic record reaches up to 8 km at its depocenter, ranging from Late Ordovician to Late Cretaceous (Fig. 2). The evolution of those units is linked to the uplift and subsidence caused by the Southern Gondwana orogeny. The Paraná Basin is subdivided into six supersequences: (i) Supersequence Rio Ivaí (Ordovician–Silurian), (ii) Supersequence Paraná (Devonian), (iii) Supersequence Gondwana I (Carboniferous–Lower

Triassic), (iv) Supersequence Gondwana II (Triassic), (v) Supersequence Gondwana III (Jurassic–Cretaceous) and (vi) Supersequence Bauru (Cretaceous) (Zalan et al., 1990; Milani, 1997; Milani and Ramos, 1998; Cagliari et al., 2014) (Fig. 1 and 2).
The Supersequence Gondwana I (Fig. 2) has a transgressive-regressive trend and a gradual paleoenvironmental change, from glacially influenced environments at the bottom to arid environments at the top (Milani, 1997). Sedimentary record begins

with the glacial deposits of Itararé Group (Fig. 2). At the top of this group, glacial conditions associated to marine transgressions were deposited (Palermo and Rio Bonito formations) (Lavina and Lopes, 1987). The Rio Bonito Formation is composed of estuarine and deltaic deposits at its base, and facies of barrier-beach with lagoon bodies and marshes associated (including thick coal layers). The progressive relative sea level rise resulted in shoreface deposits (upper part of Rio Bonito Formation) and offshore of Palermo Formation (Fig. 2). The adjacent unit (Irati Formation) is composed by siliciclastic

deposits with organic matter rich intervals, carbonate and evaporites (Lavina, 1992). The Irati Formation is overlain by the Serra Alta and Teresina Formations, which represent the establishment of continental environments in the basin. The top of this supersequence (Rio do Rasto and Pirambóia Formations) represent non-marine sedimentation, with lacustrine, fluvial and eolian facies indicating the establishment of arid conditions along the Mesozoic (Lavina, 1992).
The Irati Formation is Artinskian (278.4 ± 2.2 Ma) (Santos et al., 2006) composed of fine siliciclastic sediments,

occasionally rich in organic matter, intercalated with carbonates and evaporites. The formation is subdivided into two members: Taquaral (lower) and Assistência (upper). The Taquaral member is composed of mudstones and siltstones, while the Assistência member has siltstones and black shales, occasionally intercalated with carbonates (Mendes et al., 1966; Padula, 1968; Petri and Coimbra, 1982; Oelofsen and Araújo, 1983; Lavina, 1992; Hachiro, 1996; Milani et al., 1997; Araújo, 2001). The Irati Formation was deposited in marine restricted epicontinental environment, with salinity fluctuations,

substrate oxygenation and bioproductivity (Lavina, 1992; Hachiro, 1996; Araújo, 2001; Rodrigues et al., 2010a; Rodrigues et



al., 2010b). It is a carbonate ramp system, divided into external, middle and inner ramps (Araújo, 2001), and this association represents deposition during marine influx restriction that operated in the Paraná Basin (Lavina, 1992; Milani et al., 1997). The black shales and carbonates of the Assistência Member were studied, showing high values of TOC up to 22% in shales, and the intercalated carbonate has low values of insoluble residues (Padula, 1968; Araújo, 2001; Rodrigues et al., 2010a; Rodrigues et al., 2010b; Alferes et al., 2011). The Assistência Member is also characterized by the occurrence of skeletal remains of *Mesosaurus*, correlated to the Whitehill Formation in southern Africa (Oelofsen and Araújo, 1983; Oelofsen and Araújo, 1987; Lavina, 1992; Soares, 2003).

## 4 Results and interpretations

### 4.1 Depositional System

Core description resulted in nine different facies. The facies succession shows the occurrence of a carbonate ramp (Burchette and Wright, 1992; Wright and Burchette, 1996; Bosence and Wilson, 2003; Miall, 2010). This ramp is divided into outer, middle and inner domains, all recorded in the deposits here studied.

### 4.1.1 Outer ramp (OR)

For the studied deposits, the following facies were described: OR1) Black shale rich in organic matter (Fig. 4a), presenting high peaks of TOC (around 20%), very high total S (1 to 3.5%) and $R_{ins}$ around 75% (Fig. 5); OR2) argillaceous dark gray siltstone with fissility, sometimes medium gray and with incipient lamination (Fig. 4b). In addition, it possesses rare fossiliferous levels with fish scales in its upper part (Fig. 4f). The values of TOC are lower than 1% with local increases, the total S (%) is low and the $R_{ins}$ is high (85%) (Fig. 5); OR3) Siltstone is light gray, with sparse wavy beds composed of very fine sandstone (Fig. 4c). It presents very low values of both TOC (%) and S (%), and $R_{ins}$ (%) values are very high (Fig. 5); OR4) Heterolithic unit composed of mudstone and very fine sandstone; here dark gray mudstone is found interbedded with milimetric to centimetric (0.1 to 2 cm) lenses of fine sandstone (Fig. 4d). The contact between mudstone and sandstone is abrupt, and mudstone layers are truncated by sandstone layers. Contacts between sandstone and mudstone, on the other hand, are either abrupt or gradational, showing predominantly low TOC (%) values with local increase, with oscillatory pattern of S (%) and high values of $R_{ins}$ (%) (Fig. 5). In some specific levels skeletal remains of *Mesosaurus* are present (Fig. 4g). In 22 out of 23 cores, there are ribs, vertebrae, limb bones and teeth of mesosaurids (Cassel and Lavina, 2013; Cassel and Lavina, 2014; Cassel et al, 2016).

Physical processes involved in this domain were characterized by predominance of decantation of fine siliciclastic sediments and low energy. Yet, the influence of distal oscillatory flows and reworking currents (bottom or storms) were identified, and are well evidenced in the OR4 facies due to bone fragments disarticulation (Cassel and Lavina, 2013). These oscillatory flows are also shown by the low saw-pattern of the TOC (%), S (%) and high saw-pattern of the $R_{ins}$ (%). There were moments with anoxia conditions shown by TOC (%) high values, even euxinia due the concomitant light increase of $R_{ins}$ (%) (explained in the subsequent subitem 4.2) occured in the OR1 facies.

The outer ramp domain is located right below the storm waves interaction zone, where the deposition is dominated by fine siliciclastic sediments, where anoxia may occur (Burchette and Wright, 1992; Wright and Burchette, 1996; Bosence and Wilson, 2003; Miall, 2010).

### 4.1.2 Middle ramp (MR)

The MR1 facies occurs in two different levels that are characterized by the intercalation between carbonate and siliciclastic sedimentation. This intercalation occurs through the cyclic gradation from mudstone to carbonate. In the middle portion of this facies, a tepee level was observed (Fig. 4e) and salt layer similar to anhydrite nodules (Fig 4e). Above this interval there



are low angle laminated carbonates, usually showing wavy lamination at the top (wave ripples and hummocky crossed stratification). The intercalated siliciclastic intervals are composed of mudstone, occasionally with layers of very fine sandstone. Low TOC (%) is present, with peaks oscillating toward higher values but not exceeding 2%. S (%) is variable but predominantly low, while $R_{ins}$ (%) has a saw-pattern between 15% and 85% (Fig. 5).

The physical processes in this domain demonstrate the action of oscillatory fluxes and reworking currents (either bottom or storm). The cyclic gradation from mudstone to carbonate corresponds to shallowing pulses (Fig. 4e). The structures described above (possible anhydrite nodules and mainly the most intense tepee) correspond to shallowing evidence. They were also grouped at the middle ramp due to low thickness and occasional occurrence. Since ramp morphology possesses a smooth declivity, ephemeral sea level oscillations may occur, causing the exposition of this domain, or its flooding (which
explains the oscillatory values of TOC, % and $R_{ins}$ %).

### 4.1.3 Inner ramp (IR)

The IR1 facies is a carbonatic breccia containing centimetric angular shapped clasts (Fig. 4e). These clasts present a dome microrelief lamination and V-shaped molds. In this facies, the values of TOC (%) and $R_{ins}$ (%) are low whereas S (%) are high (Fig. 5).

Clasts lamination is characteristic of microbialites (Araújo, 2001). Breccias and V-shaped molds are evidence of subsequent aerial exposition. Geochemical data show the presence of low organic matter (low TOC, %), high content of carbonate (high $R_{ins}$, %) with higher salinity (S,%) This evidence corresponds to the inner ramp, which is a shallow water environment where carbonatic sedimentation takes place, subjected to subaerial exposition.

### 4.2 Organic geochemistry and spectral gamma-ray

### 4.2.1 Organic matter

TOC (%) values of Irati Formation have two aspects; (i) organic matter-rich horizons (> 1%) (Arthur and Sageman, 1994) distributed along the succession; and (ii) two anomalous TOC (%) occurrences of high peaks (around 20%) (Fig. 5).

Kerogen was characterized as type II, and subordinately as type III, through HI and OI data (Tissot and Welte, 1984) (Fig. 6). No abundance was registered of organic matter-rich lipidic fractions with H (the most reactive fraction) to reach Type I
levels.

The HI/TOC graphic demonstrates an asymptotic pattern, where the increasing input of organic matter is deprived of the corresponding increase in HI (Fig. 6). This asymptotic point (HI=300 and COT=4) belongs to the bituminous shale facies (OR1) (Fig. 4a). Palynological data indicate that the highest amount of amorphous organic matter of the interval, with significant content of H (Araújo, 2001), occurs from this TOC level onwards (3% to 20%). In these siliciclastic organogenic
layers, the aforementioned data are associated with high values of S (%) and peaks of reduction of $R_{ins}$ (%) (increase in the carbonate level) uncommon inside the bituminous shales (%) (Fig. 5).

Points characterized as type III (HI<100) and transitional II-III (Fig. 6) occur in lower values of TOC (<2%) (Fig. 5), in fine siliciclastic, non organogenic facies (OR2, OR3 and OR4).

The low content of H in organic matter which resulted in type II and III means that the organic matter either has experienced
degradation or it has marine origin (Type II) and subordinately continental (Type III), in a mixture or alternated pattern.

The asymptotic pattern between HI and TOC (%) in the organogenic facies shows variations of precursor source, or delimitates the beginning of selective biodegradation. The amorphus organic matter analyzed in these intervals demonstrates that the kerogen came from H-rich sources (Araújo, 2001). Thereby these organogenic intervals from OR1 facies contain an organic matter H-rich partially biodegraded, resulting in a kerogen fairly poor in H (Type II). High values of S (%)
corroborate these data. High S (%) indicates partial aerobic biodegradation of organic matter in its movement along the oxic water column associated with selective biodegradation of elements more metabolizable (H) by sulphate reducing bacteria.





At specific levels from the organogenic facies OR1, euxinic environment was identified, because of peaks of reduction of $R_{ins}$ (%). With dissolved sulphate available in water, the metabolic activity of bacteria promotes a sulphate reduction using organic matter and generating bicarbonate and sulphydric acid as byproduct, characterizing euxinic environments. The process of sulphate reduction raises the pH and causes the precipitation of calcium carbonate. Data from biomarkers and

isotopes confirm the bacterial influence in organic matter (Rodrigues et al., 2010a; Rodrigues et al., 2010b; Alferes et al., 2011).

The points characterized as type III (HI<100) and transitional II-III (Fig. 6) are non-organogenic strata, where low HI indicates mixture of different sources (marine and continental). Confirmation of this trend occurs by predominance of allochthonous palynomorphs (Araújo, 2001) and the correspondence to chemostratigraphic units which indicate marine and

terrestrial organic matter according to biomarkers (Rodrigues et al., 2010a; Rodrigues et al., 2010b; Alferes et al., 2011). These data also indicate the increase of marine organic matter in relation to continental near to the maximum flooding surface of Irati A Sequence.

### 4.2.2 Oxygenation

One of the proxies about oxygenation is the presence of organic matter. Organic matter is conditioned to the level of

oxygenation of the substrate, since preservation occurs under low levels. The eU (ppm) values of spectral gamma ray is a proxy to identify less oxygenation beyond the levels enriched in organic matter, due to precipitation of authigenic eU (ppm). Besides the peaks of increase in the eU (ppm) curve in the organogenic facies, the authigenic U (ppm) occurs distributed along the studied section (Fig. 7). The anoxic levels are also seen in the eTh/eU curves where values near or below 2 indicate anoxia (Wingall and Twitchett, 1996). These curves reinforce the negative covariation between U (ppm) and Th (ppm) (in

the form of monazite, a resistate detritic mineral that depends of allochthonous influx) (Fig. 7) (Andersson and Worden, 2004).

These data show the occurrence of anoxia. These conditions highlight in the organogenic facies, and are also demonstrated by the authigenic eU along the studied section. These data demonstrate the recurrence of discontinuous moments of anoxic conditions along the studied section.

### 25     4.2.3 Salinity

S (%) data provide information regarding organic matter, and also salinity. In the carbonatic strata, high values of S (3.5 %) occur associated to low levels of TOC (%) and $R_{ins}$ (%) (Fig. 5). Related to these carbonatic strata, salt layers occur with texture similar to anhydrite nodules. Next to these levels a slight decrease in the eTh (ppm) occurs. Besides these specific levels the eTh/eU ratio presents values between 2 and 7 predominantly along the succession in the siliciclastic non

organogenic facies. There are isolated levels where the eTh/eU ratio presents high values (Fig. 7).

The high values of S (%) in carbonatic stata are in accordance with the presence of sulphate (gypsite and anhydrite), typical of evaporitic IR domain. The S (%), which has large disponility as sulphate dissolved in sea water, becomes enriched in the inner part of the ramp, as described here in association with IR1 facies and in thin and isolated levels at the MR1 facies. The salt layer with texture similar to anhydrite nodules reinforces the aforementioned data. These strata are correlated to

carbonates associated with marine organic matter and hypersalinity conditions, according to isotope and biomarkers (Rodrigues et al., 2010a; Rodrigues et al., 2010b Alferes et al., 2011). The eTh (ppm) data corroborates the evaporitic conditions, demonstrating the low humidity according to the reduction of continental detritic minerals input.

These carbonatic levels with hipersalinity and low humidity conditions are reported in the carbonatic non organogenic facies IR1 and MR1. Also, the data from eTh/eU ratio predominant in the succession define an oxic marine environment for the

siliciclastic non organogenic facies, but with sparce influence of oxic continental input (Wingall and Twitchett, 1996). Studies in correlated strata based in biomarkers show normal salinity with continental contribution for this siliciclastic non





organogenic facies of the remaining succession and normal salinity for the organogenic facies (Rodrigues et al., 2010a; Rodrigues et al., 2010b Alferes et al., 2011).

The data show two moments of high salinity and low humidity in the carbonatic facies IR1 and IR2. For the rest of succession, the siliciclastic facies has normal salinity, with sparse continental input in the siliciclastic non organogenic
facies.

### 4.3 Stratigraphic scheme of depositional sequences

Three depositional sequences were identified in the studied interval, as previously mentioned by Araújo (2001) and Hachiro (1996). However, a detailed register of the system tracts is here proposed.

The Irati A Sequence begins with a transgressive system tract (TST) characterized by the succession of the facies OR3 and
OR2 (Fig. 4), with reduction of grain size representing relative sea-level rise (Fig. 5). The subtle inflexion of the TOC (%) curve towards higher values is in accordance with this trend, because it indicates either reduction in the oxygenation close to the substrate or reduction in the siliciclastic input (Fig. 5). This interval has normal salinity with sparse continental input according to correlated studies (Rodrigues et al., 2010a; Rodrigues et al., 2010b; Alferes et al., 2011). The maximum flooding surface (MFS) of this tract is highlighted by local increase of TOC (%) and S (%). In this surface, previous studies
showed increment of marine organic matter. The highstand system tract (HST) overlies the TST and is characterized by grain size increase in facies OR4, which corresponds to a progradational trend still with normal salinity (Fig. 5). Reduction in the accommodation space associated to the smooth morphology of the ramp facilitated the subaerial exposition even during the short relative sea level variations in this sector of the ramp. In some cases, it caused pedogenetic processes.

On top of this first sequence, where the IR occurs, the forced regression system tract (FRST) is recorded, where sea-level
lowering allowed the development of biotic activity typical of this type of ramp (Fig. 5). These microbialites are indicated by low $R_{ins}$ (%) demonstrating a high level of carbonates and low TOC (%), and suitable oxygenation (Fig. 5). This interval has higher salinity and low humidity. The abrupt contact of this level with the subjacent facies characterizes an erosive surface of marine regression (Fig. 5). The subsequent formation of breccias in these deposits demonstrates a relative sea level fall which caused subaerial exposition characterizing, therefore, a sequence boundary (SB) (Fig. 5). Such boundary is identified
in all cores described from the area. The isolated occurrence of bioturbation is characteristic of consolidated substrate excavation, which reinforces this interpretation (Buatois et al., 2002).

The Irati B Sequence begins with a lowstand system tract (LST) characterized by the occurrence of MR domain (Fig. 4e), where repetitive pulses of shallowing. These pulses are limited by adjacent flooding surfaces and are interpreted as lowstand parasequences (Fig. 5). This pattern is in accordance with the $R_{ins}$ (%) curve which oscillates significantly, and some short
and isolated peaks of increase in TOC (%) (Fig. 5). This interval has higher salinity and low humidity conditions. At the top of this level, the presence of sedimentary evidence of oscillatory fluxes and reworking currents, as well as the increase in siliciclastic sediments, indicates relative sea-level rise, which occurred at the end of the lowstand system tract (Fig. 5). The overlying transgressive system tract (TST) culminates at black shales from facies OR1 (Fig. 4a), which has a flooding surface that is identified in gamma-ray logs and geochemical indicators, especially TOC between 16% and 20% (Fig. 5). As
exposed in the previous subitem, this interval has normal salinity and the organic matter has bacterial influence. This relative sea-level means the reestablishment of the OR (Fig. 5). The subsequent highstand system tract (HST) is characterized by the reduction of organic matter content in the black shales, seen in the TOC (%) values for facies OR2 and normal salinity (Fig. 5). As observed in the previous sequence the contact with the IR domain is abrupt and establishes an erosive surface of marine regression (Fig. 5). The forced regression system tract (FRST) is characterized by the occurrence of bioherms and
under higher salinity and low humidity conditions. Subsequently, the IR domain suffered subaerial exposition resulting in breccias, characterizing a sequence boundary. In the Irati B Sequence this system tract is less expressive, thinner and absent in some cores.



The Irati C Sequence possesses lowstand system tract (LST) at the base, characterized by the occurrence of the MR domain under higher salinity and low humidity conditions, however thinner than the previous sequence. Its carbonatic sedimentation is evidenced in the $R_{ins}$ (%) curve (Fig. 5). The adjacent sea-level rise characterizes the transgressive system tract (TST). This tract is demonstrated by the succession of OR4 and OR1 facies (Fig. 4d and 4a), recovering normal salinity, with

predominance of the outer ramp, besides the tendency of TOC (%) values to rise (Fig. 5). The reduction in the amount of organic matter in the subsequent unit (Serra Alta Formation) characterizes the highstand system tract (HST) (Fig. 5).

### 4.4 High frequency cycles

Cyclic transgressive-regressive (TR) internal oscillations were observed along the three depositional sequences (Fig. 7) according to Embry and Johannessen (1993).

The total amount of gamma-ray is defined predominantly by the K (%), due to its relative abundance in relation to the other radioactive elements measured. However, it is influenced by the eU (ppm) and the eTh (ppm), as negative covariations, reducing the effect of K (%) in the total curve, as well as positive covariation, which reinforce the effect of K (%) in the curve. These fenomena are seen in the studied section logs.

Positive covariation between K (%) and eU (ppm) was observed in eight out of nine analyzed cycles in the succession (Fig.

7), influenced predominantly by K (%) variation. Due to the affinity of eU (ppm) with low oxygenation levels, such covariation suggests oxygen depletion in transgressive trends. The increase in peak of K (%) with a negative covariation of eU (ppm) in cycle 8 demonstrates a rise in the relative sea-level (Fig. 7), without the establishment of low oxygenation context.

There is also significant increase in the eU (ppm) values that do not coincide with K (%) peaks and, therefore, are not

conspicuous in the total curve (Fig. 7). These levels, which indicate low oxygenation, are not related with sea level rise. In this resolution, no covariation is observed between eU (ppm) with TOC (%) to justify the oxygen depletion by events of productivity increase. Therefore, the peaks of authigenic U (ppm) indicate climatic changes which influenced the hydric imbalance and the circulation patterns next to the substrate.

### 5 Final considerations

The depositional system of the Irati Formation is interpreted as a carbonatic ramp with occurrence of three domains: inner, middle and outer. The succession of the domains corresponds to three depositional sequences with TST, HSST, FRST and LSST. The depositional dynamics is complex and variable, since it is influenced not only by oceanic incursions, but also by continental input through climate.

In relation to high frequency cycles, T-R cycles were observed, identified in oscillations of K (%) values, highlighted by the

increase of eU (ppm). This indicates lower oxygenation near the substrate, according to the transgressive trend. Besides the T-R cycles, the peaks of eU (ppm) indicate phases of lower oxygenation beyond the transgressive trend. In these cases of high frequency oscillation, the absence of relation with TOC (%) indicates climate as the controlling factor through its influence in the hydric imbalance and circulation pattern.

Along the studied section, there are variations in the salinity, oxigenation of the substrate and in organic matter composition.

There is predominance of normal salinity in the fine siliciclastic facies of the OR both in the organogenic (OR1) and in non-organogenic ones. In the carbonatic facies of the IR and MR, there is evidence of high salinity and low humidity. Anoxic conditions were identified in recurrent discontinuous moments along the studied section intercalated with oxic levels, besides the two intervals of anoxia even euxinia in the siciliclastic organogenic facies (OR1). Organic matter is of marine origin in the carbonatic facies of the IR and the MR. In the fine siliciclastic non-organogenic facies there are predominance of marine



organic matter, however with continental input. The precursors of the organic matter are of bacterian origin in the organogenic facies (OR1).

This indicates different conditions of hydric imbalance. An evaporitic environment in a semi-arid climate was established in two moments of the sucession, with low humidity and poor continental contribution. Such conditions are demonstrated in the carbonatic levels from IR and MR domains, associated to the RFST and LST, where microbial carbonatic sedimentation took place under oxic conditions, in an environment of high salinity and subaerial exposition. On the other hand, there was the reestablishment of normal salinity with humidity restored in the OR domain associated to a sea level rise in TST and HST. Highlighted anoxia level and even euxinia occurred in the OR1 facies. This paleoenvironmental context resulted from the relative sea level rise (evidenced by the increase of marine organic matter next to SIM) and the return of humidity conditions (evidenced by the light increace in the eTh curve and contribution of terrestrial organic matter). The summation of these factors culminated in the deposition of the organogenic facies due water column stratification through sea level rise and increase of the bioproductivity promoted by nutrient input from the continental.

Climate influence on the deposition can be noted through the following evidence: a) Oscilation between low and high humidity conditions evidenced by low continental input through decrease of detrital minerals (eTh (%) data), that resulted in hipersalinity and semi-arid conditions in the carbonatic facies IR1 and MR1; b) The case of oxygen depletion as evidenced by eU (ppm) outside the transgressive peak from high frequency cycles. In this case, climate influenced the hydric balance between continental input and marine influx. This promoted increase or reduction of bottom circulation due to the intermittent presence of thermohaline circulation, which reduces the oxygen supply downwards to the substrate. C) In the case of T-R cycles evidenced by K (%) the climatic influence caused the increase in thickness of water column during transgressions in this scale.

On the other hand the eustatic influence is noted only in the depositional sequence hierarchy. Near the maximum flooding surfaces from the depositional sequences, a reduction of continental input occurred. This is shown by increase of marine organic matter, and demonstrates the predominance of allochthonous control (oceanic influx) in the sea level oscillation in this sector of the basin in relation to the autochthonous control (continental contribution).

Two factors controlled the deposition of sediments (allochthonus and autochthonus), and two different hierarchies are observed in this control. One hierarchy is the activity of climate in the high frequency cycles. This was seen through the authigenic eUppm along the studied section outside the transgressive peak from high frequency cycles. This demonstrates the recurrence of discontinuous moments of ephemeral anoxic conditions caused by the hydric balance between the continental input and the marine influx. The other higher hierarchy is in the depositional sequences where eustasy acts in addition to climate at a different scale. Influence of eustasy is observed through the increase of marine organic matter next to the SIM. The climate at another scale is observed through the oscilation between low and high humidity highlighted in the carbonate facies (MR and IR) and siliciclastic organogenic facies (OR1). Previous studies referred to climatic control of facies succession in the Irati Formation, ascribing a 4[th] order hierarchy to the ciclicity here observed in the three depositional sequences (Araújo, 2010; Hachiro, 1996). Climate is the major controller of sequence deposition in the 4[th] order scale (Araújo, 2010; Hachiro, 1996). Besides, we infer that the cycles of highest frequency are of 5[th] order hierarchy, and are mostly influenced by climate.

## 6 Conclusion

We explain the depositional process from Irati Formation. From this, we affirm that these processes are strongly associated to the catastrophic Permian geodynamic. Anoxia, euxinia, organic-rich shales with anomalous TOC and anomalous carbonate occurrence is also result of the same Permian extinction triggers. Which has models that propose a coordinated



imposition of physical perturbations that are linked (e. g. bolide impact, Siberian flood basalt volcanism, anoxia in the oceans and release of seafloor methane).

**Acknowledgments**

The authors thank all the comments, suggestions and criticism made by Gerson Terra, Paulo Sérgio Gomes Paim that helped us improve the manuscript, and the laboratory support of Lauro Moreira da Rosa and Laís Vieira de Souza. Thanks to the Thin Section Laboratory of UNISINOS University and the Chemostratigraphy and Organic Geochemistry Laboratory at the Faculty of Geology in Rio de Janeiro State University for the analysis, and the Geological Survey of Brazil (CPRM) for the data availability and for financial support. Marlise Colling Cassel is grateful to Coordenação de Aperfeiçoamento de Pessoal de Nível Superior for their partial financial support of her master of science studies.

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



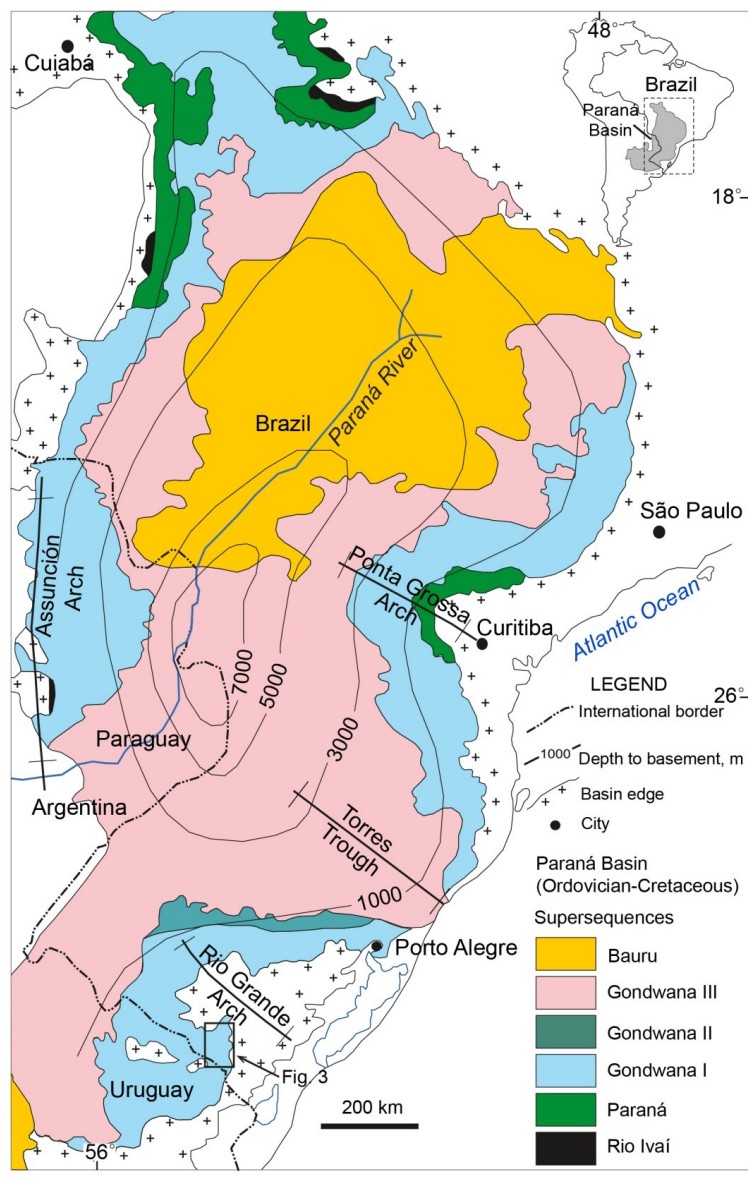

**Figure 1: Simplified geological map of the Paraná Basin (Ordovician - Cretaceous), located in the cratonic interior of South America. Contour lines show depth to basement. The colored areas represents the supersequencies that compose the basin fill (adapted from Milani, 1997). Inset shows location of Paraná Basin in South America.**





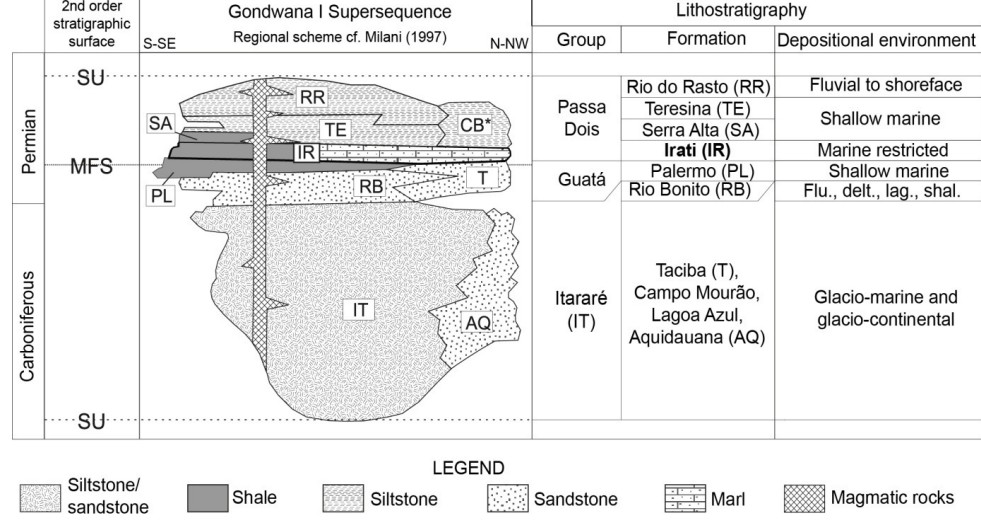

**Figure 2: Stratigraphic diagram of Gondwana I Supersequence, showing 2nd order stratigraphic surfaces SU (subaereal unconformity) and MFS (maximum flooding surface) (adapted from Milani, 1997). Irati Formation (IR) in Permian (278.4 ± 2.2 Ma) (Santos et al., 2006) is shown in bold font, composed by fine siliciclastic and carbonatic rocks. Irati Formation is above**
5 **transitional deposits of Rio Bonito Formation (RB) and below marine deposits of Palermo Formation (PL). *CB – Corumbataí Formation, with age equivalent to Serra Alta and Teresina Formations. Depositional environment: flu. = fluvial; delt. = deltaic; lag. = lagoonal; shal. = shallow marine.**



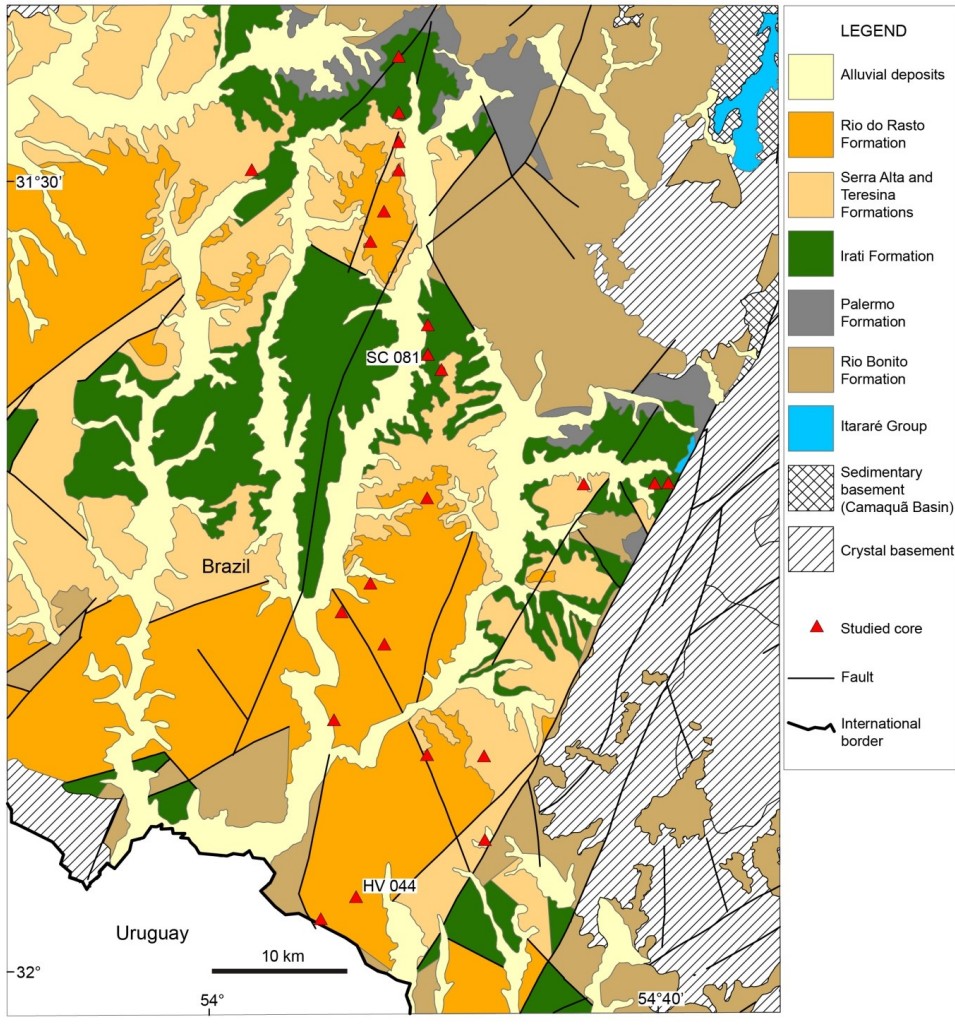

**Figure 3: Location of the study area in southern Brazil and 23 cores described. Colored areas are lithoestratigraphic units and cores with geochemical analysis are indicated (HV-044, SC-081).**





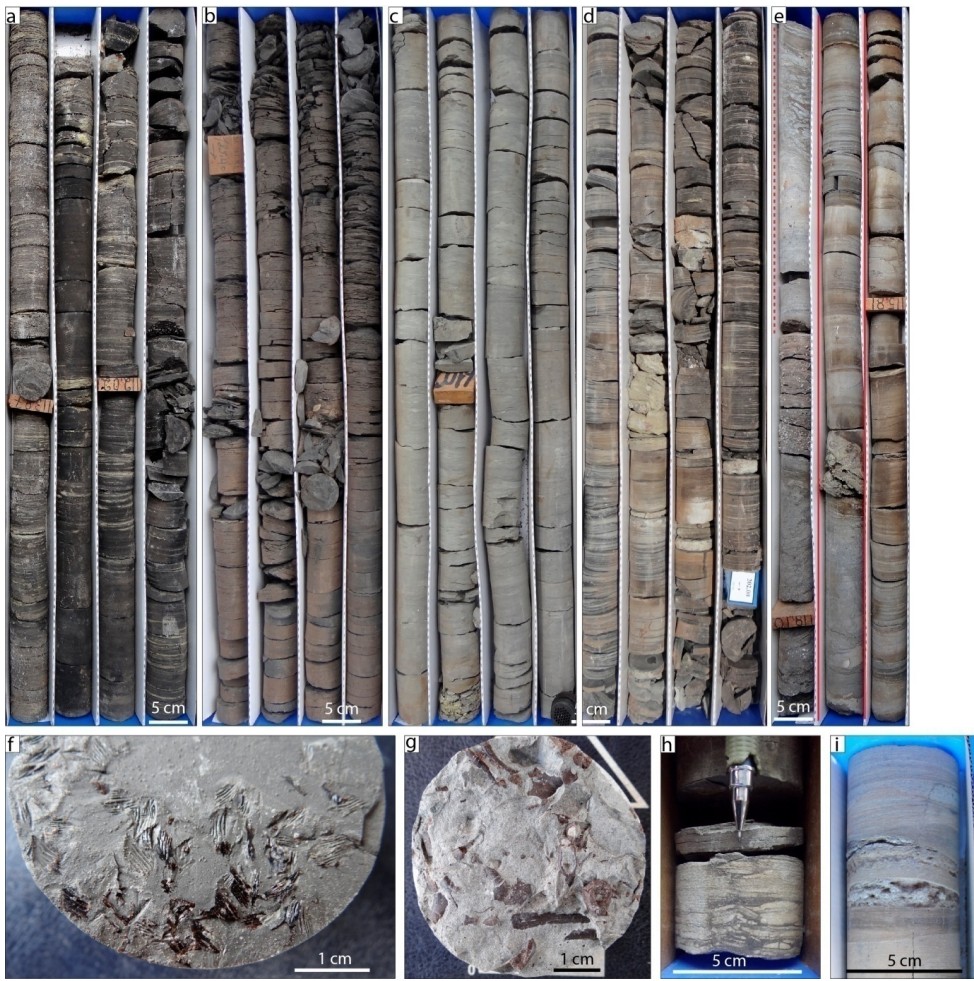

**Figure 4: Photos showing facies variation divided into outer ramp (OR), middle ramp (MR) and inner ramp (IR) domains. a) Facies OR1 - black shales. b) Facies OR2 - dark gray argillaceous-siltstone with fissility and incipient lamination. c) Facies OR3 - light gray siltstone. d) Facies OR4 - heterolithic. e) The IR level is indicated by dashed red line, overlain by MR in continuous red line. f) Fossiliferous level with fish scales of OR2 facies. g) Fossiliferous level with Mesosuarus remains of OR4 facies. h) Tepee of MR shallowing. i) Salt layer similar to anhydrite nodules in MR.**





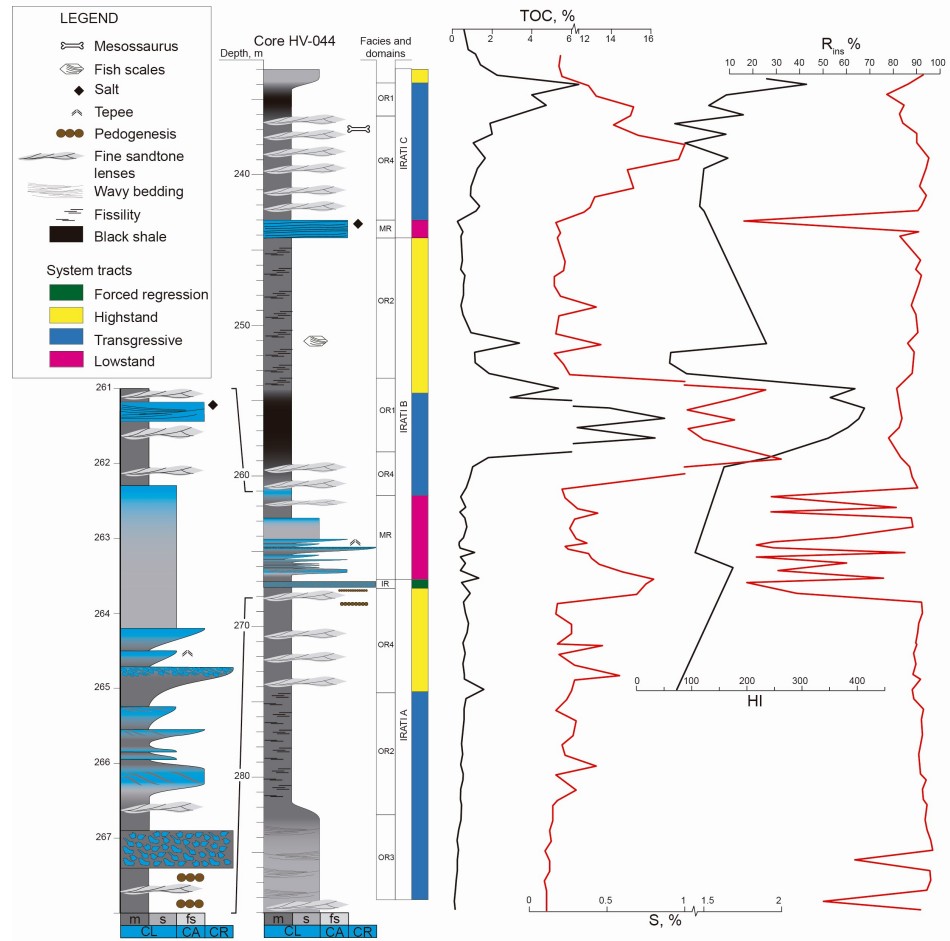

**Figure 5: Lithologic and stratigraphic profile from the described core HV-044, selected to display the data. The other core with the same analyses SC-081 is in the attached files. The figure shows the depositional sequences (IRATI A, IRATI B, IRATI C), their system tracts, the domains of the carbonatic ramp (IR = inner ramp; MR = middle ramp and OR = outer ramp) and the curves resulting from the geochemical analyses of the total organic carbon (TOC, %), sulphur (S, %), hydrogen index (HI) and insoluble residue (R$_{ins}$,%). In the core HV-044 we postulated the pedogenesis and salt levels based in their presence in all the ther core descripted (e.g. SC-081 in the attached files).**



**Figure 6: Correlation graphics of geochemical data. a) Correlation of oxygen index (OI) and hydrogen index (HI) of HV-044 and their resulting kerogen compositional fields (II). b) Graphic correlation of total organic carbon (TOC, %) and IH, a standard line showing the asymptotic behavior in the cluster for HV-044. c) Correlation graphics of oxygen index (IO) and hydrogen index (IH) of SC-081 and their resulting kerogen compositional fields (II). d) Graphic correlation of total organic carbon (TOC, %) and IH, a standard line demonstrating the asymptotic behavior in the cluster for SC-081.**




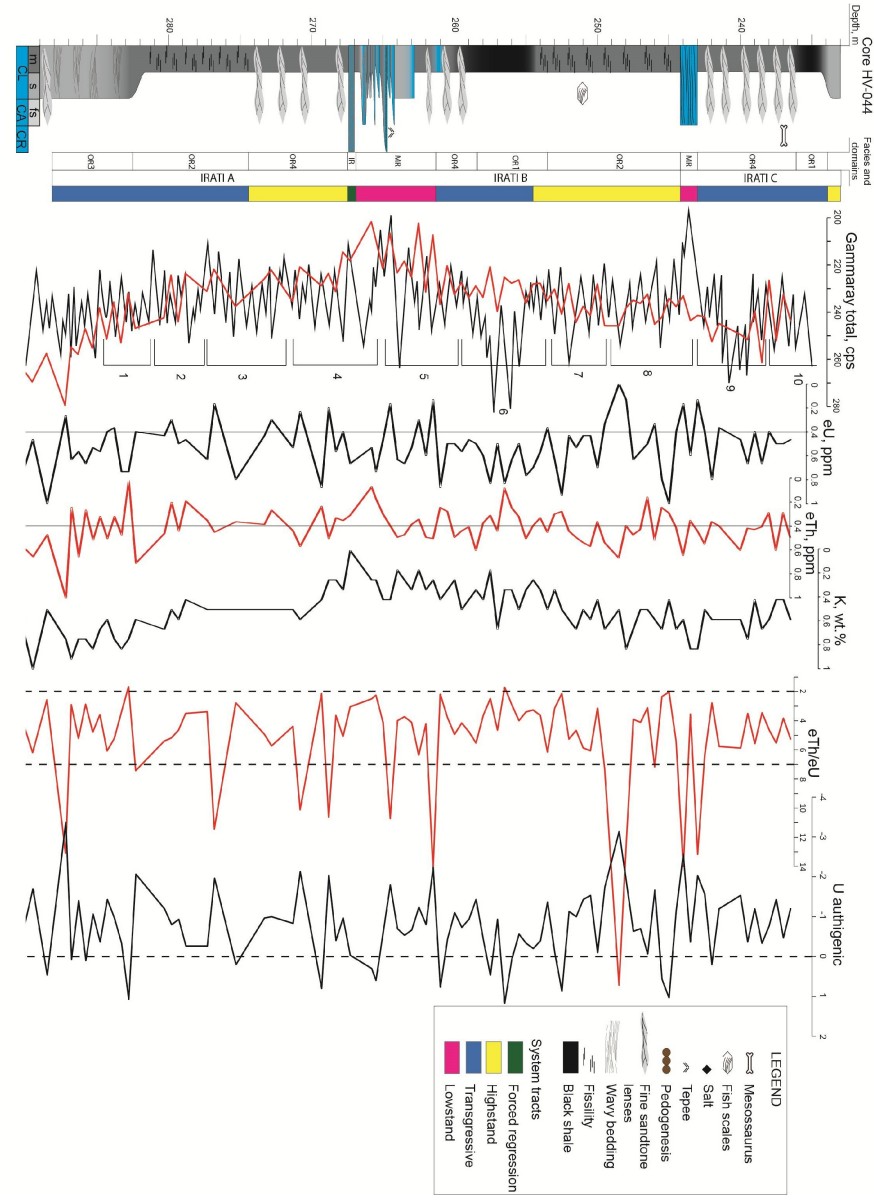

**Figure 7: Lithologic and stratigraphic profile of the described core HV-044, which we selected to demonstrate the data. The figure show the depositional sequences (IRATI A, IRATI B, IRATI C), their system tracts, the domains of the carbonatic ramp (IR = inner ramp; MR = middle ramp and OR = outer ramp), the total amount curves of gamma-ray and the differentiable channels from uranium (eU, ppm), thorium (eTh, ppm) and potassium (K, %), and derivative ratios (eTh/eU, U authigenic). In the gamma-ray total curves, the red one was sampled in this study and the black one is from CPRM. Numbers next to the gamma-ray total curves are the T-R cycles (e.g., 1 = 1$^{th}$ cycle). In the U authigenic curve, the dashed line delimitates the occurrence of its precipitation. In the eTh/eU curve, the dashed line highlights the values close to 2 and 7. Values near or below 2 indicates anoxia, whereas values between 2 and 7 indicate oxyc marine environment and higher values indicate oxic continental input (Wingall and Twitchett, 1996).**