# Peer review of "Anoxia and salinity changes: a new Permian catastrophe record"

_Climate of the Past, 2019_

## Referee Comment (RC1) · Anonymous Referee #1 · 12 Jun 2019

The paper by Marlise Cassel and co-authors is a comprehensive study of the Irati formation in southern Brazil, with a complex carbonate ramp history derived from numerous sources and proxies. There is a wealth of data in this paper, and the interpretations of the depositional history of the Paraná basin are based on grounded arguments. It is therefore my view that this dataset should eventually be published in a journal such as Climate of the Past. However, in its current form, the manuscript is in need of substantial work prior to eventual publication.

Firstly, the level of English language and grammar is well short of what is required of an international journal. There are numerous spelling mistakes throughout (e.g. Wingall instead of Wignall, hiper instead of hyper), and the current structure and use of abbreviations makes the manuscript extremely difficult to read. It took me 4 afternoons

to get through the paper completely. If it is that much of a slog for someone who has agreed to review, then it will fail to be read by much of its target audience. I understand that it must be difficult to prepare a manuscript in a language other than one's native language, but the level of mistakes are too many to even begin suggesting corrections. The manuscript needs to be checked by someone with English as a 1st language to improve readability and grammar. I would also try and cut down on the number of abbreviations (e.g. Outer Ramp etc.) because it does nothing to help readability and the paper is not limited by a page count.

The second major issue with the paper in its current form is the catastrophe focus of the title, abstract, and interpretation. The end-Permian is indeed the largest mass extinction of the Phanerozoic, but the early- to mid-Permian ($\sim$pre 270 Ma) was not typified by the mass extinction of genera. Neither the Emeishan ($\sim$259 Ma) nor Siberian Traps ($\sim$251 Ma) are likely to be coincident with the Irati Fm, and these are among the prime contenders for ecosystem stress that led to the end-Permian mass extinctions. It is difficult to orientate when exactly the Irati formation is in Figure 2, given the lack of available dates, but it would appear that the majority of shales are found in the early Permian strata. I am not saying that this data is not interesting, it is just not indicative of mass extinctions. As much of the abstract, introduction, and conclusions frame this work in this context, all of these sections need to be rewritten. The structure of the paper needs to be altered accordingly, as the main points of this manuscript are tracking sea level changes and the response of shelf environments to these changes.

Thirdly, there is an urgent need for a graphical illustration of the Stratigraphic Scheme of depositional sequences (section 4.3). This would significantly aid the reader in understanding how the system evolved from inner ramp to outer ramp facies, and what this means for the evolution of the Paraná basin.

Overall, I recommend major revisions. There is are good dataset here, but it is currently marred by poor English and false linkage to end-Permian environmental disturbances.

Minor comments: Figure 1. The boundaries of Paraguay, Uruguay, and Argentina are incomplete. Figure 2. If there is any available age constraints on the Irati Fm., please add them. Figure 3. Again, providing even rough ages of the different lithostratigraphic units would be very valuable here.

––––––––––––––––––––––––––––––––

---

## Referee Comment (RC2) · Anonymous Referee #2 · 14 Jun 2019

This paper by Cassel et al. focuses on the early Permian Irati Formation in Brazil. Authors try to determine and constrain potential anoxic episodes in several cores from this basin, and to link these events to other major known bio or geological Permian events known elsewhere. At a first view, the used dataset and analyses appear numerous and some interpretations valid. Some obtained information are maybe also important although local. Overall, I was highly disconcerted by the form of the text and the discussion/conclusion included in this work. In my opinion, the paper requires numerous significant modifications and additions. To be brief:

1- I fully understand that it is very difficult for some authors to write in English, but this text is not at all up to the level of a standard publication. There are too many mistakes and typos pervading the ms (thus I did not list them hereafter). It took me a very

long time to read it (several times) and (try to) to understand ideas contained in many paragraphs and sentences. It is necessary that authors have their text corrected by a native-English speaker because, as it stands, very few passages are simply understandable by the reader.

2- Title, discussion and conclusion focus on major Permian extinction events, such as during the latest Permian, and link them to major geological events, such as the Siberian traps. However, the Irati Formation is not coeval to any of these events and thus, this link made by the authors is not understandable. None of the data shown in this work can be linked to any mass extinction. This is interesting to document that the Irati Formation potentially records anoxic conditions but they are local or regional and should be discussed with coeval events worldwide (if identified). Authors should also provide more biostratigraphical data allowing to constrain the age of this formation. Overall, the entire structure of the paper has to be changed.

3- Authors claimed that 23 cores were studied but data from only two sites are presented. Authors should provide data from these other sites with their corresponding environments as the reader can follow interpreted environmental fluctuations in space and time.

4- Authors should also provide true facies analyses with their corresponding interpretation in terms of depositional environments by providing pictures of thin sections or polished slabs: a few pictures of rock colors from one of the studied core is not enough to check authors' claims about depositional environments and anoxia, and throughout the entire basin. Also, authors should provide pictures of bioturbation or bioherms when they discussed them in the text. It can help the reader to follow authors' interpretations.

Overall, mainly based on the poor quality of the English and of the false links made between observed local/regional environmental changes and Permian extinctions, requiring a complete rewriting of the paper, I recommend to reject the paper.

---

## Author Comment (AC1) · 4 Aug 2019

To:

Yves Godderis

Climate of the Past Editor August 4th, 2019.

**SUBJECT:** Revision of manuscript "Anoxia and salinity changes: a new Permian catastrophe record".

Dear Dr. Godderis,

Below are the final author comments (AC) in response to Anonymous Referee #1 comments about our manuscript, which we would like you to consider for publication in Climate of the Past.

Best regards.

Marlise Colling Cassel

Note 1: Referees comments are in "red", whereas our answers are in black.

**Note 2:** Many minor corrections suggested by the reviewers have been directly included in the revised text (see the attached document where changes to the original version have been highlighted).

**Comments from the Anonymous Referee #1:**

The paper by Marlise Cassel and co-authors is a comprehensive study of the Irati formation in southern Brazil, with a complex carbonate ramp history derived from numerous sources and proxies. There is a wealth of data in this paper, and the interpretations of the depositional history of the Paraná basin are based on grounded arguments. It is therefore my view that this dataset should eventually be published in a journal such as Climate of the Past. However, in its current form, the manuscript is in need of substantial work prior to eventual publication.

Firstly, the level of English language and grammar is well short of what is required of an international journal. There are numerous spelling mistakes throughout (e.g. Wingall instead of Wignall, hiper instead of hyper), and the current structure and use of abbreviations makes the manuscript extremely difficult to read. It took me 4 afternoons to get through the paper completely.

If it is that much of a slog for someone who has agreed to review, then it will fail to be read by much of its target audience. I understand that it must be difficult to prepare a manuscript in a language other than one's native language, but the level of mistakes are too many to even begin suggesting corrections. The manuscript needs to be checked by someone with English as a 1st language to improve readability and grammar. I would also try and cut down on the number of abbreviations (e.g. Outer Ramp etc.) because it does nothing to help readability and the paper is not limited by a page count.

AC: The spelling mistakes flagged here have been repaired as well the excessive abbreviations. The manuscript is currently with a native English proofreader.

The second major issue with the paper in its current form is the catastrophe focus of the title, abstract, and interpretation. The end Permian is indeed the largest mass extinction of the Phanerozoic, but the early- to mid-Permian (pre 270 Ma) was not typified by the mass extinction of genera. Neither the Emeishan (259 Ma) nor Siberian Traps (251 Ma) are likely to be coincident with the Irati Fm, and these are among the prime contenders for ecosystem stress that led to the end-Permian mass extinctions.

AC: The mass extinction that occurred at the end of the Permian is the most severe biotic crisis in Earth's history. Numerous papers report that this event was responsible for decimating more than 90% of marine species and about 70% of continental vertebrate families (Erwin, 1994; Retallack, 1995; Knoll et al., 1996; Knol et al., 2007) . The pattern of disappearance of these species is quite complex, some species disappeared before and others after the Permo-Triassic (P-T) limit. Permian cause-of-death models demonstrate that the extinction of individuals occurred due to hypercapnia, hypoxia and oceanic acidification due to generalized anoxic and euxinic conditions, possibly to the photic zone (Clapham and Payne, 2011; Grice et al., 2005; Hotinski et al., 2001). These cause-of-death models can be framed in Permian-Triassic Superanoxic Event (Grice et al., 2005) or Superanoxia (Isozaki, 1994).

Many triggers were proposed for extinction at the end of the Permian. In addition to the Siberian and Emeishan traps, geologists have also focused on ocean anoxia and catastrophic methane release from the seabed (Knoll et al., 2007, Grice et al., 2005). Thus, contrary to what has been described in the past, the extinction of the PT would not have been caused by a single mechanism, but rather the interrelation of several factors: (i) the important marine regression, which reduced marine habitats and increased climatic variability; (ii) anoxia in the oceans and catastrophic release of methane on the seabed (Cassel, 2017; Knoll et al., 2007, Grice et al., 2005); (iii) global climate change; (iv) the intense volcanism in Siberia (Reichow et al., 2009); and (v) meteor impact (Becker et al., 2001).

The end of the Permian was an extreme state of the Earth's surface system. It was characterized by maximum continental aggregation and minimum marine flooding of continents. There was also an unusual east-west ocean basin in the tropics and, after the melting of the Gondwana and Kazanian glaciers in Angara, a low temperature gradient from the equator to the pole. Such conditions profoundly influenced the course of events between the Permian and Triassic (Knoll et al., 2007).

The low temperature gradient from the equator to the pole leads to poor ocean circulation and widespread anoxia in the oceans. Polar warming and tropical cooling of sea surface temperatures cause anoxia throughout the deep ocean as a result of lower dissolved oxygen in lower source waters and increased nutrient utilization. Accumulation of H2S and CO2 in the oceans is sufficient to directly cause mass extinction (Hotinski et al. 2001).

Therefore anoxia plays an important role in driving extinction. Superficial overturn of sulfuric waters and emissions of hydrogen sulfide into the atmosphere provide a killing mechanism that can explain terrestrial and marine extinctions. The association of oxygen-poor water and rapid transgression or overturn of deepwater is critical to the hypothesis that anoxia caused the extinction event (Grice et al., 2005).

Some authors point out that anoxia occurred in much shallower waters than previously recognized and show that sections of high latitude boreal oceans were also affected by the anoxic event. (Wingnall and Twitchett).

Some models indicate that the Permian extinction, as well as the Neoproterozoic, occurred through the repeated occurrence of anomalous conditions. Thus, such a model predicts that high-resolution stratigraphy within basins will show rapid pulsed extinctions, with the latest clustered appearances associated with C isotopic excursions and other evidence for anoxic deep ocean turnover (Knoll et al, 1996; Knoll et al, 2007). This model is in line with Wignall and Twitchett (1996) observation that Permian extinction can be a considerably more complex event, involving an initial extinction separated from the most recent mass slaughter of the late Permian.

Therefore, it was not a single extinction event that occurred at ~ 252.3 My, but rather extinction pulses (Song et al., 2013). In addition, environmental (and catastrophic) stress events would have preceded the major extinction (Knoll et al., 2007; Ward et al., 2005). An example is the extinction that occurred about 10 Myr before the great extinction, preserved in the geological record between the Guadalupian and the Lopingian (Stanley and Yang, 1994). Already in Kungurian there are radical episodes of anoxia. These are explained by climate trends between the tropics and Gondwanna, related to the Late Paleozoic Icehouse-greenhouse transition. This phenomenon caused oceanic stagnation and consequent anoxia (Liu et al 2017).

Thus, the Irati Formation presents these anomalous characteristics typical of the Permian catastrophic environmental crises and is characterized as another record of this singular period in the Earth history. Such a relation had not yet been reported in this way, especially once in Gondwana.

It is difficult to orientate when exactly the Irati formation is in Figure 2, given the lack of available dates, but it would appear that the majority of shales are found in the early Permian strata.

AC: The figure has been repaired, leaving the dated levels explicit to better locate the Irati Formation interval in time. However, it is important to explain that a paper is underway, with high precision dating TIMS method, that will make the Irati Formation even younger.

I am not saying that this data is not interesting, it is just not indicative of mass extinctions. As much of the abstract, introduction, and conclusions frame this work in this context, all of these sections need to be rewritten. The structure of the paper needs to be altered accordingly, as the main points of this manuscript are tracking sea level changes and the response of shelf environments to these changes.

AC: These sea level changes and the responso f shelf envoronments are also relevant to an paper. But it could be another paper and for a journal with another proposal. In this case, we do relate the Irati Formation to the end-Permian disasters that culminate in the P-T extinction. This relation is explicit in the title, but perhaps not sufficiently clarified throughout the abstract, introduction and conclusions. But we hope to have better explained this relationship by answering this point in the previous paragraphs.

Thirdly, there is an urgent need for a graphical illustration of the Stratigraphic Scheme of depositional sequences (section 4.3). This would significantly aid the reader in understanding how the system evolved from inner ramp to outer ramp facies, and what this means for the evolution of the Paraná basin.

AC: This missing figure was made and added to the paper.

Overall, I recommend major revisions. There is are good dataset here, but it is currently marred by poor English and false linkage to end-Permian environmental disturbances.

AC: We greatly appreciate the review and suggestions we accepted. As previously written, the text is already is currently with a native English proofreader, and we hope to have explained more clearly that this is not a false linkage between the Irati Formation and the Permian catastrophes, which are triggers for the extinction.

Minor comments: Figure 1. The boundaries of Paraguay, Uruguay, and Argentina are incomplete. Figure 2. If there is any available age constraints on the Irati Fm., please add them. Figure 3. Again, providing even rough ages of the different lithostratigraphic units would be very valuable here.

AC: We cite the ages in all figures that mention any stratigraphic units.

Below are the figures with the modifications made and new figures added as requested by the referee.

---

## Author Comment (AC2) · 4 Aug 2019

To:

Yves Godderis

Climate of the Past Editor                                        August 4[th], 2019.

**SUBJECT:** Revision of manuscript "Anoxia and salinity changes: a new Permian catastrophe record*".*

Dear Dr. Godderis,

Below are the final author comments (AC) in response to Anonymous Referee #2 comments about our manuscript, which we would like you to consider for publication in Climate of the Past.

Best regards.

Marlise Colling Cassel

**Note 1:** Referees comments are in **"red"**, whereas our answers are in **black.**

**Note 2:** Many minor corrections suggested by the reviewers have been directly included in the revised text (see the attached document where changes to the original version have been highlighted).

**Comments from the Anonymous Referee #2:**

This paper by Cassel et al. focuses on the early Permian Irati Formation in Brazil. Authors try to determine and constrain potential anoxic episodes in several cores from this basin, and to link these events to other major known bio or geological Permian events known elsewhere. At a first view, the used dataset and analyses appear numerous and some interpretations valid. Some obtained information are maybe also important although local. Overall, I was highly disconcerted by the form of the text and the discussion/ conclusion included in this work. In my opinion, the paper requires numerous significant modifications and additions. To be brief:

1- I fully understand that it is very difficult for some authors to write in English, but this text is not at all up to the level of a standard publication. There are too many mistakes and typos pervading the ms (thus I did not list them hereafter). It took me a very long time to read it (several times) and (try

to) to understand ideas contained in many paragraphs and sentences. It is necessary that authors have their text corrected by a native-English speaker because, as it stands, very few passages are simply understandable by the reader.

AC: The manuscript is currently with a native English proofreader.

Regarding the distribution of data and whether they are regionally representative, we have made an additional figure that demonstrates the regional extent of the described facies (see Figure 4 at the end of the document).

2- Title, discussion and conclusion focus on major Permian extinction events, such as during the latest Permian, and link them to major geological events, such as the Siberian traps. However, the Irati Formation is not coeval to any of these events and thus, this link made by the authors is not understandable. None of the data shown in this work can be linked to any mass extinction. This is interesting to document that the Irati Formation potentially records anoxic conditions but they are local or regional and should be discussed with coeval events worldwide (if identified). Authors should also provide more biostratigraphical data allowing to constrain the age of this formation. Overall, the entire structure of the paper has to be changed.

We do not link to siberian traps. There are other trigger possibilities for of the Permian extinction beyond the Siberian traps. There pulses of environmental disasters causing stress to the previous P-T boundary. The data brought here may instead be linked to these catastrophic Permian episodes, which in turn may have caused the deaths. The data shown in this study, as answered in the previous item, are of regional magnitude. Additional dating data were cited in all figures that mention any stratigraphic units.

The mass extinction that occurred at the end of the Permian is the most severe biotic crisis in Earth's history. Numerous papers report that this event was responsible for decimating more than 90% of marine species and about 70% of continental vertebrate families (Erwin, 1994; Retallack, 1995; Knoll et al., 1996; Knol et al., 2007) . The pattern of disappearance of these species is quite complex, some species disappeared before and others after the Permo-Triassic (P-T) limit. Permian cause-of-death models demonstrate that the extinction of individuals occurred due to hypercapnia, hypoxia and oceanic acidification due to generalized anoxic and euxinic conditions, possibly to the photic zone (Clapham and Payne, 2011; Grice et al., 2005; Hotinski et al., 2001). These cause-of-death models can be framed in Permian-Triassic Superanoxic Event (Grice et al., 2005) or Superanoxia (Isozaki, 1994).

Many triggers were proposed for extinction at the end of the Permian. In addition to the Siberian and Emeishan traps, geologists have also focused on ocean anoxia and catastrophic methane release from the seabed (Knoll et al., 2007, Grice et al., 2005). Thus, contrary to what has been

described in the past, the extinction of the PT would not have been caused by a single mechanism, but rather the interrelation of several factors: (i) the important marine regression, which reduced marine habitats and increased climatic variability; (ii) anoxia in the oceans and catastrophic release of methane on the seabed (Cassel, 2017; Knoll et al., 2007, Grice et al., 2005); (iii) global climate change; (iv) the intense volcanism in Siberia (Reichow et al., 2009); and (v) meteor impact (Becker et al., 2001).

The end of the Permian was an extreme state of the Earth's surface system. It was characterized by maximum continental aggregation and minimum marine flooding of continents. There was also an unusual east-west ocean basin in the tropics and, after the melting of the Gondwana and Kazanian glaciers in Angara, a low temperature gradient from the equator to the pole. Such conditions profoundly influenced the course of events between the Permian and Triassic (Knoll et al., 2007).

The low temperature gradient from the equator to the pole leads to poor ocean circulation and widespread anoxia in the oceans. Polar warming and tropical cooling of sea surface temperatures cause anoxia throughout the deep ocean as a result of lower dissolved oxygen in lower source waters and increased nutrient utilization. Accumulation of $H_2S$ and $CO_2$ in the oceans is sufficient to directly cause mass extinction (Hotinski et al. 2001).

Therefore anoxia plays an important role in driving extinction. Superficial overturn of sulfuric waters and emissions of hydrogen sulfide into the atmosphere provide a killing mechanism that can explain terrestrial and marine extinctions. The association of oxygen-poor water and rapid transgression or overturn of deepwater is critical to the hypothesis that anoxia caused the extinction event (Grice et al., 2005).

Some authors point out that anoxia occurred in much shallower waters than previously recognized and show that sections of high latitude boreal oceans were also affected by the anoxic event. (Wingnall and Twitchett).

Some models indicate that the Permian extinction, as well as the Neoproterozoic, occurred through the repeated occurrence of anomalous conditions. Thus, such a model predicts that high-resolution stratigraphy within basins will show rapid pulsed extinctions, with the latest clustered appearances associated with C isotopic excursions and other evidence for anoxic deep ocean turnover (Knoll et al, 1996; Knoll et al , 2007). This model is in line with Wignall and Twitchett (1996) observation that Permian extinction can be a considerably more complex event, involving an initial extinction separated from the most recent mass slaughter of the late Permian.

Therefore, it was not a single extinction event that occurred at ~ 252.3 My, but rather extinction pulses (Song et al., 2013). In addition, environmental (and catastrophic) stress events would have preceded the major extinction ( Knoll et al., 2007; Ward et al., 2005). An example is the extinction

that occurred about 10 Myr before the great extinction, preserved in the geological record between the Guadalupian and the Lopingian (Stanley and Yang, 1994). Already in Kungurian there are radical episodes of anoxia. These are explained by climate trends between the tropics and Gondwanna, related to the Late Paleozoic Icehouse-greenhouse transition. This phenomenon caused oceanic stagnation and consequent anoxia (Liu et al 2017).

Thus, the Irati Formation presents these anomalous characteristics typical of the Permian catastrophic environmental crises and is characterized as another record of this singular period in the Earth history. Such a relation had not yet been reported in this way, especially once in Gondwana.

3- Authors claimed that 23 cores were studied but data from only two sites are presented. Authors should provide data from these other sites with their corresponding environments as the reader can follow interpreted environmental fluctuations in space and time.

AC: To answer this request, additional figures were made to clarify the correlation of the described facies in all cores (see Figure 4 at the end of the document). To answer the request about flutuations in space and time see additional Figures 5 and 7 at the end of the document).

4- Authors should also provide true facies analyses with their corresponding interpretation in terms of depositional environments by providing pictures of thin sections or polished slabs: a few pictures of rock colors from one of the studied core is not enough to check authors' claims about depositional environments and anoxia, and throughout the entire basin. Also, authors should provide pictures of bioturbation or bioherms when they discussed them in the text. It can help the reader to follow authors' interpretations.

AC: To answer this request we made an additional figure showing the facies variation according to sedimentary domain as well as sedimentary product and mechanism. In this figure new photos of facies, including thin sections, and bioturbation were added (see additional Figures 5 at the end of the document).

Overall, mainly based on the poor quality of the English and of the false links made between observed local/regional environmental changes and Permian extinctions, requiring a complete rewriting of the paper, I recommend to reject the paper.

AC: We greatly appreciate the review and suggestions we accepted. As previously written, the text is already is currently with a native English proofreader, and we hope to have explained more clearly that this is not a false linkage between the Irati Formation and the Permian catastrophes, which are triggers for the extinction.

Below are the figures with the modifications made and new figures added as requested by the referee.

[Figure]

Figure 2: a) Stratigraphic diagram of Gondwana I Supersequence, showing 2[nd] order stratigraphic surfaces SU (subaereal unconformity) and MFS (maximum flooding surface) (adapted from Milani, 1997). The Irati Formation (IR) in the Permian (278.4 ± 2.2 Ma) (Santos et al., 2006) is shown in bold font, composed by fine siliciclastic and carbonatic rocks. The Irati Formation is above the transitional deposits of Rio Bonito Formation (RB) and below the marine deposits of Palermo Formation (PL). *CB – Corumbataí Formation, with age equivalent to Serra Alta and Teresina Formations. Depositional environment: flu. = fluvial; delt. = deltaic; lag. = lagoonal; shal. = shallow marine. b) Stratigraphic column of Irati Formation, showing Taquaral and Assistência Members with previous dating: (1) Santos et al., 2006; (2) Rocha-Campos et al., 2019.

[Figure]

**Figure 3: Location of the study area in southern Brazil, the 23 cores described and a composite section. Colored areas are lithoestratigraphic units, its ages and the cores with geochemical analysis are indicated (HV-044, SC-081).**

[Figure]

**Figure 4: The composite section shows the correlation of the described facies. The correlation extends regionally, as shown by Rodrigues et al. (2010a) data. These data are from another core located in the Paraná State, 1000 km north to the studied area, and shown the chemostratigraphic unit (A-G) of Rodrigues et al, (2010a), directly correlated with the facies described in this study. TOC = total organic carbon, S = total sulphur and R$_{ins}$ = insoluble residue.**

[Figure]

**Figure 5: Photos showing facies variation divided into outer ramp (OR), middle ramp (MR) and inner ramp (IR) domains. a) Facies OR1 - black shales. b) Facies OR2 - dark gray argillaceous-siltstone with fissility and incipient lamination. c) Facies**

OR3 - light gray siltstone. d) Facies OR4 - heterolithic. e) A detail of fish scales of OR2. f) A detail of Mesosaurus skeletal remains of OR4. g) Low angle laminated carbonates of Middle Ramp Domain. h) The cyclic gradational intercalation from mudstone to carbonate of Middle Ramp Domain. i) Salt layer similar to anhydrite nodules in the Middle Ramp Domain. j) Tepee in the Middle Ramp Domain. k) Carbonatic breccia containing centimetric angular shaped clasts of Inner Ramp Domain. l) Bioturbation level of the Inner Ramp, which is characteristic of consolidated substrate excavation. m and n) Thin sections of the clasts showing a dome microrelief lamination and V-shaped moulds decribed by Araujo, 2001.

[Figure]

**Figure 7: The stratigraphic scheme of the Irati Formation depositional sequences: IRATI A, IRATI B and IRATI C. The figure shows the stratigraphic column with the facies stacking and sedimentary domains of a carbonatic ramp (IR = inner ramp; MR = middle ramp and OR = outer ramp) (see Figure 5), and the system tracts.**